# Prebiotic condensation through wet–dry cycling regulated by deliquescence

Thomas D. Campbell [1,3], Rio Febrian [1,3], Jack T. McCarthy [1], Holly E. Kleinschmidt [1], Jay G. Forsythe [2] & Paul J. Bracher [1]

Wet–dry cycling is widely regarded as a means of driving condensation reactions under prebiotic conditions to generate mixtures of prospective biopolymers. A criticism of this model is its reliance on unpredictable rehydration events, like rainstorms. Here, we report the ability of deliquescent minerals to mediate the oligomerization of glycine during iterative wet–dry cycles. The reaction mixtures evaporate to dryness at high temperatures and spontaneously reacquire water vapor to form aqueous solutions at low temperatures. Deliquescent mixtures can foster yields of oligomerization over ten-fold higher than non-deliquescent controls. The deliquescent mixtures tightly regulate their moisture content, which is crucial, as too little water precludes dissolution of the reactants while too much water favors hydrolysis over condensation. The model also suggests a potential reason why life evolved to favor the enrichment of potassium: so living systems could acquire and retain sufficient water to serve as a solvent for biochemical reactions.

---

[1] Department of Chemistry, Saint Louis University, 3501 Laclede Avenue, St. Louis, Missouri 63103, USA. [2] Department of Chemistry and Biochemistry, College of Charleston, 66 George Street, Charleston, South Carolina 29424, USA. [3]These authors contributed equally: Thomas D. Campbell, Rio Febrian. Correspondence and requests for materials should be addressed to P.J.B. (email: paul.bracher@slu.edu)

Elucidating the means by which the first functional biopolymers arose on Earth is a major focus of origin-of-life research[1–3]. Condensation reactions—like the conversion of amino acids into peptides—present a considerable challenge, because the reactions are thermodynamically disfavored in water[4,5]. Wet–dry cycling is commonly viewed as a feasible means of driving condensation reactions in prebiotic conditions to generate mixtures of prospective protobiopolymers, including peptides, depsipeptides, nucleic acids, and others[6–16]. The standard model for wet–dry cycling invokes iterative phases of cool temperatures and rain followed by periods that are hot and dry. The hot and dry phase of each cycle drives the condensation reactions, while the wet phase supplies solvent to permit better diffusion of reactants than is possible in solid mixtures[17,18]. One obvious shortcoming of the model is its reliance on rainstorms or flooding as controlled sources of water. While flooding and run-off are sometimes invoked as advantages in prebiotic scenarios[19], the challenge of overdilution to cellular compartmentalization is a well-known problem in the origin of life[20,21]. Before the existence of enclosed cells protected by membranes or other structures, it is difficult to imagine flooding as anything, but predominantly destructive to protobiotic reaction mixtures.

Deliquescent substances—which form aqueous solutions by absorbing water vapor—offer a solution to the overdilution problem. Deliquescent salts absorb a limited amount of water from the air, based on the relative humidity. A mixture of deliquescent salts would regulate the volume of the system as modern cells have evolved to do[21,22].

A substance is deliquescent if it spontaneously acquires water vapor from its surrounding atmosphere—at a given temperature and relative humidity (%RH)—to form a homogeneous aqueous solution[23,24]. As the humidity increases around a potentially deliquescent substance, water vapor adsorbs to the surface until the humidity reaches the critical deliquescent point ($RH_0$), where spontaneous dissolution occurs[23,24]. At $RH_0$, the vapor pressure of the newly formed solution is less than the partial pressure of the water vapor in the atmosphere. Mass transfer from the atmosphere to the sample occurs until an equilibrium is reached. The reverse process, efflorescence, occurs when the %RH value is lower than $RH_0$[23]. When a substance effloresces, water evaporates from the mixture and the solute crystallizes. $RH_0$ values vary from one substance to another, and when two or more substances are in physical contact or a mixture, deliquescence lowering can occur[24]. For example, a mixture of KCl and NaCl deliquesces at lower %RH than either of the individual salts[25].

Deliquescent compounds have been identified in natural geological settings, including systems where deliquescent salts play a role in enabling aqueous liquid mixtures to exist in environments that are otherwise too cold and/or dry to support liquid water. The Don Juan Pond in Antarctica is adjacent to several steep-sloped water tracks, which are rich in deliquescent $CaCl_2$. In summertime, meltwaters running through the tracks wash calcium-rich brine into the pond, augmenting water levels that fluctuate seasonally in a body of water that rarely freezes, despite surface temperatures that typically descend to $-50\,°C$ in winters[26]. In a hyperarid region of Chile's Atacama Desert that is otherwise inhospitable to life, endolithic microbial communities reside in deposits of halite. Their photosynthetic activity spikes when the relative humidity rises above 70%, exceeding the threshold $RH_0$ where their halite environment becomes deliquescent[27]. Deliquescent mixtures of chlorides and perchlorates have been identified on Mars[28]. These mixtures appear to flow seasonally and have garnered significant interest from astrobiologists as the only extant liquid environments on the surface of the planet[29].

Motivated by reports of natural deliquescent mixtures existing as liquid solutions in arid environments on Earth and Mars—and recognizing how hydration by deliquescence offers an alternative to the contrived reliance on perfect storms to deliver limited amounts of water at regular intervals in standard wet–dry cycling experiments—we sought to test: (i) whether deliquescence could be an effective method for regulated rehydration of a prebiotically relevant reaction mixture subjected to wet–dry cycling, (ii) whether deliquescent mixtures could serve as effective hosts for cycled condensation reactions, and (iii) whether hydration regulated by deliquescence is advantageous relative to manual addition of water that simulates rain events in typical prebiotic experiments.

Here, we report the ability of mixtures of deliquescent minerals to serve as media for the condensation of amino acids into polypeptides during self-regulated, iterative wet–dry cycling. These reaction mixtures evaporate to dryness at high temperatures and reacquire water vapor from the atmosphere to form aqueous solutions at low temperatures, thereby rehydrating without the addition of water by a rainstorm and avoiding the possibility of destructive overdilution. Ostensibly minor differences—e.g., in ambient humidity or the replacement of $K^+$ with $Na^+$ counterions—can lead to profound differences in the propensity of samples to absorb water, and hence, large differences in the yields of condensation reactions they host.

## Results

**Selection of cycling conditions.** When choosing conditions for model prebiotic experiments, it is necessary to balance historical plausibility with experimental convenience. Although an Earth day was shorter at 4 Ga[30], we selected cycles that lasted 24 h as a reasonable approximation that was also convenient to monitor over many days. The hot-and-dry phases were set to 100 °C or 120 °C, temperatures in line with those found in previous studies[31,32]. While several previous studies relied on longer drying times[33]—which generally produce higher yields of condensation—long periods of heat (>6 h) seem unlikely given the 24-h diurnal cycle on Earth. We therefore limited our heating periods to 4–6 h per cycle.

For the cool phase of each cycle, the relative humidity (%RH) was varied from 30–70 %RH. Given the wide range of %RH values on modern Earth—from single digits in desert climates to 100% when raining—these values are both reasonable in a historical context and convenient, as they are easily simulated in the laboratory. Values of %RH much higher or lower are considerably more difficult for a humidity chamber to maintain with precision.

We maintained the temperature at 40 °C for the cool phase of the wet–dry cycles. A temperature slightly elevated from room temperature was necessary to ensure that a constant temperature, and relative humidity could be maintained despite daily fluctuations in the ambient conditions of the laboratory, e.g., a particularly warm or humid day. The higher cool temperature seems relevant to early Earth, as there is geological evidence it was generally warmer than today[34].

**Selection of model reaction and substrates.** We targeted the oligomerization of amino acids into peptides as a model condensation reaction for our study for its obvious relevance to prebiotic chemistry. Glycine is the simplest and most prebiotically relevant amino acid. It is produced in the highest yields in simulated prebiotic syntheses of amino acids[2,35–37], and a multifactor analysis by Trifonov suggests that glycine was the most prevalent amino acid in the earliest proteins[38]. As a matter of experimental rigor, yields of oligoglycines can be measured quantitatively by liquid chromatography[14,39]. Glycine is the only

canonical amino acid that produces achiral oligomers, which simplifies their analysis relative to other amino acids. Chromatographic analysis is crucial because the high quantity of salts present greatly complicates quantitative analysis by mass spectrometry (see Supplementary Fig. 30, Supplementary Table 25, and related discussion in the Supplementary Discussion). The results from our experiments are summarized in Supplementary Tables 1 and 2.

**Deliquescence regulates reversible and limited hydration**. We first examined the hydration of a simple mixture of glycine with $K^+$, $Na^+$, $Cl^-$, and $OH^-$ salts. Chloride salts are the water-soluble minerals most abundant on Earth[40]. Hydroxide was selected as a simple source of base to catalyze condensation, which will proceed less efficiently near neutral pH[14]. While neither KCl nor NaCl are deliquescent by themselves at 70 %RH, a mixture of the two and their hydroxides lowered the critical relative humidity ($RH_0$) to the point where the mixture was deliquescent. The reaction mixture used for the initial cycling experiments had a 20:10:1 molar ratio of chlorides:glycine:hydroxides with a 3:1 molar ratio of $Na^+$:$K^+$ (see Supplementary Methods and Supplementary Figs. 2 and 3).

Identical samples of 50 mg of the salt mixture in glass scintillation vials were cycled with cooling phases at 40 °C and three different relative humidities: 30, 50, and 70 %RH. The samples at 70 %RH—a value above the critical $RH_0$ of the mixture—collected and lost ~0.05 g of water per cycle, enough to create homogeneous aqueous solutions during the cool phases. The samples at 30 and 50 %RH—below the critical RH—varied by <0.01 g and remained white solids throughout both the hot and cool phases of ten complete cycles. Figure 1 shows the change in mass (corresponding to moisture loss and uptake) over time for the 30 and 70 %RH samples, demonstrating not only how deliquescence (dependent on the $RH_0$ value) can have a profound influence on the rehydration of identical samples, but also on the precise control of this rehydration in terms of reproducibility and reversibility.

**Effects of humidity on yields of oligomerization**. Next, we tested the ability of these mixtures of salts to host the condensation of glycine by wet–dry cycling with rehydration mediated by changes in atmospheric humidity. Samples were subjected to ten full 24-h cycles of heating (at 100 °C or 120 °C) and cooling (at 30, 50, or 70 %RH and 40 °C). For samples subjected to 100 °C, the heating phases lasted 6 h and the cooling phases lasted 18 h. For samples subjected to 120 °C, the heating phases lasted 4 h and the cooling phases lasted 20 h.

Figure 2 shows the total yield of glycine oligomers ($Gly_{\geq 2}$) for the six combinations of heating and cooling conditions. Regardless of the temperature of heating, the samples that were deliquescent and formed aqueous solutions during the cooling phase (at 70 %RH) produced substantially higher yields than those that were not deliquescent and remained solid throughout the experiment (at 30 and 50 %RH). All samples that did not rehydrate due to insufficient humidity for deliquescence produced only limited yields of oligomers, below 2.5%. At 70 %RH, where the mixtures were deliquescent, those subjected to heating at 120 °C produced higher total yields of oligomer after a lesser number of cycles, but by the end of the experiment (ten cycles), the samples heated at 100 °C were comparable. Supplementary Figs. 4–9 and Supplementary Tables 4–9 show histograms that track the yields of each oligomer measured at the end of cycles 1, 2, 3, 5, and 10. There is a clear advantage to periodic rehydration of these samples (at 70 %RH) versus remaining dry throughout (at 30 and 50 %RH).

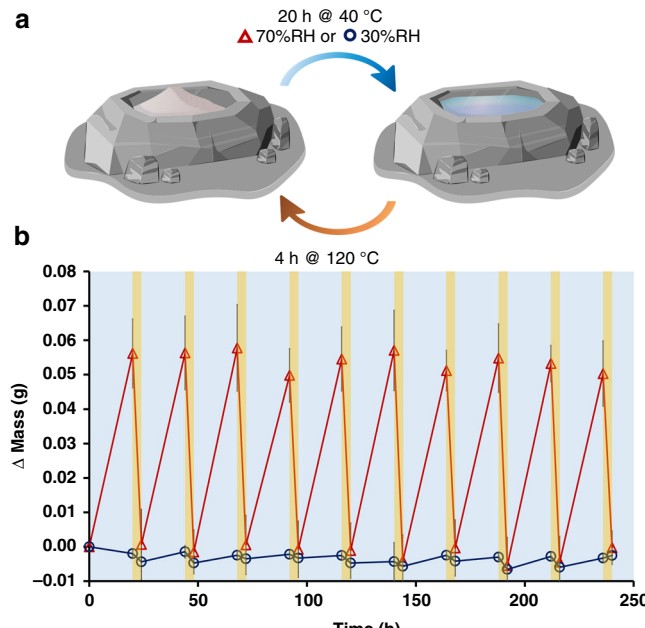

**Fig. 1** Demonstration of deliquescent wet–dry cycles. **a** An illustration of our deliquescent model with its cycling conditions. **b** A demonstration of the magnitude, reversibility, and consistency of moisture sorption/desorption during wet–dry cycling of a 50 mg mixture of $K^+$, $Na^+$, $Cl^-$, $OH^-$, and glycine. Containers were weighed over the course of ten complete 24-h cycles to monitor moisture gain and loss at 70 %RH (red triangles, above $RH_0$ for the mixture) and 30 %RH (blue circles, below $RH_0$ for the mixture). Each cycle consisted of two phases: a cool phase for 20 h at 40 °C (blue background) followed by a hot phase for 4 h at 120 °C (orange background). The error bars represent 95% confidence intervals ($n = 3$ identical experiments). Source data are provided as a Source Data file

In a control experiment, we tested glycine oligomerization starting from homogeneous solutions of the chloride mixtures as opposed to dry mixtures. In these experiments, all of the sample mixtures started as wet in the first cycle before being subjected to cycling with either 30 or 70 %RH cool periods. We observed that both sets of mixtures gave comparable yields at the end of the first cycle, but the samples cooled at 70 %RH (such that every cycle had both a wet and a dry phase) gave considerably higher yields for all subsequent cycles versus those cooled at 30 %RH, which dried after the first cycle and remained dry because $RH_0$ was never surpassed (see Supplementary Figs. 12–19 and Supplementary Tables 12–17).

**Benefits of regulated rehydration**. The results above are consistent with the idea that the increased diffusion of reactants permitted in solution more than compensates for hydrolysis that occurs in the presence of the extra water[17,18]. But there is clearly a limit to the benefits of water in these systems. In an experiment designed to study the effects of overhydration, the same mixtures of glycine with $K^+$, $Na^+$, $Cl^-$, and $OH^-$ salts were subjected to phases of 6 h at 100 °C and 18 h at 40 °C and 70 %RH, but at the end of the cooling period, an additional 20 mL of deionized water was added to a selection of the samples to simulate overhydration. After ten cycles, the samples hydrated by deliquescence alone gave an average total yield of glycine oligomers of 16.0%, while the samples subjected to additional hydration by deionized water gave an average yield of 1.9% (Fig. 3). The addition of extra water better simulates what would be expected of hydration by rainstorms, which unlike rehydration by deliquescence, cannot be counted on to provide a limited amount of water. The extra water

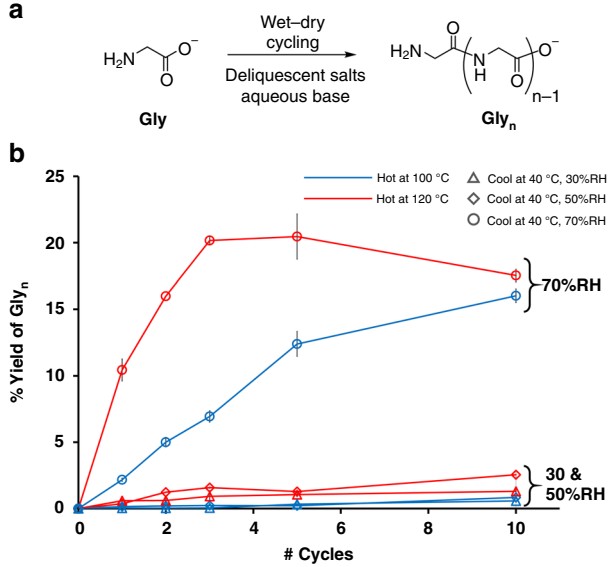

**Fig. 2** The condensation of glycine in a deliquescent system. **a** The condensation of glycine into oligoglycines. We report yields based on the percentage of initial glycine converted into oligomers, excluding cyclic DKP dimer. **b** The total yields of glycine oligomers in the presence of $(K^+/Na^+)$ $(^-Cl/^-OH)$ after 1, 2, 3, 5, and 10 cycles. Each cycle was 24 h. For the red samples, one cycle included 20 h at 40 °C and 70 %RH (circles), 50 %RH (diamonds), or 30 %RH (triangles) followed by 4 h at 120 °C. For the blue samples, one cycle included 18 h at 40 °C and 70, 50 or 30 %RH (circles, diamonds, triangles) followed by 6 h at 100 °C. The error bars represent 95% confidence intervals ($n = 3$ identical experiments). The downturn in total yield observed at higher temperature is addressed in the Supplementary Discussion. Source data are provided as a Source Data file

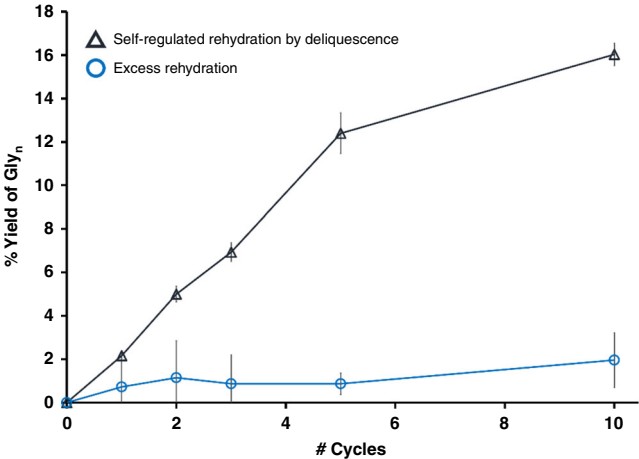

**Fig. 3** The advantage of limited rehydration in wet–dry cycling. Yields of glycine oligomers (excluding DKP) in the presence of $(K^+, Na^+)(Cl^-, OH^-)$ after 1, 2, 3, 5, and 10 cycles. Each cycle was 24 h. For the samples marked with triangles, one cycle included 18 h at 40 °C and 70 %RH, followed by 6 h at 100 °C. The samples marked with circles were exposed to the same environmental cycles, but 20 mL of water was added to the mixture before each drying period. This addition simulated heavy rain and overhydration of the sample to verify a shortcoming of the standard model for wet–dry cycling that is obviated by self-regulated, limited rehydration through deliquescence. The error bars represent 95% confidence intervals ($n = 3$ identical experiments). Source data are provided as a Source Data file

reduces yields of glycine oligomerization by increasing the duration of time the oligomers are subjected to hydrolysis and decreasing the duration of time in the cycle where condensation is favorable, after the water has evaporated. While the heat and reduced water activity during the evaporative dry phase support peptide synthesis, heat applied to an overhydrated sample—with higher water activity—can favor hydrolysis. The liquid deliquescent brine already had water activity low enough to support peptide condensation, but the evaporation of samples to dryness was critical to the growth of longer oligomers. Control samples of glycine in liquid deliquescent brines heated in capped vials—and hence, not allowed to evaporate—produced trace yields of $Gly_2$ and $Gly_3$ over 10 cycles (1.1% total), but the yields were comparatively lower than the samples allowed to evaporate to dryness (16.0% of $Gly_{\geq 2}$, with up to $Gly_{13}$ observed, see Supplementary Fig. 31).

**Deliquescence controlled by alkali salts.** Once we established the efficacy of deliquescence to rehydrate the chloride system, we sought to vary the deliquescent salts used in the mixtures to see if the results of atmospheric rehydration in wet–dry cycling are general. Phosphate salts especially piqued our interest for their importance to biology. At 50 %RH, $K_2HPO_4$ is deliquescent, while $Na_2HPO_4$ is not. While the availability of phosphate salts was likely limited on the Prebiotic Earth[41], there is no question that, at some point, phosphate anions became relevant to the development of life[42].

In this experiment, rather than vary the relative humidity, we varied the salt. Samples of glycine mixed with $K_2HPO_4$ or $Na_2HPO_4$ were subjected to ten cycles of 20 h at 40 °C and 50 % RH and 4 h at 120 °C. The results are summarized in Fig. 4. The potassium samples, which were deliquescent and rehydrated completely during the cooling periods, produced total yields of $Gly_n$ of 21.2%. The non-deliquescent sodium samples remained solid through all ten cycles and gave a total yield of $Gly_n$ of 3.5%. Figure 4b is a histogram tracking the yield of each oligomer measured at the end of cycles 1, 2, 3, 5, and 10 (see Supplementary Figs. 10, 11 and Supplementary Tables 10, 11 for more details). It is apparent that not only do the total yields improve over time for the deliquescent system, but that the distribution of products favors longer oligomers over time. For the prebiotic condensation of amino acids, this feature is presumably advantageous, as longer oligomers are better suited to developing the secondary and tertiary structure observed in modern functional proteins. Even shorter peptides may have served as functional biomolecules on Prebiotic Earth[43–45].

In control experiments designed to test whether deliquescence —and not simply potassium itself—is the critical factor responsible for the increased yields of oligomerization, we subjected glycine to wet–dry cycling with NaBr and KBr. Here, we observed significantly higher yields in the presence of NaBr versus KBr, which is consistent with the fact that NaBr is deliquescent and KBr is not at 50 %RH (see Supplementary Figs. 23–27 and Supplementary Tables 20–23). However, at 70 %RH, where both the $K^+$ and $Na^+$ reaction mixtures with glycine are deliquescent, we observed similar yields. Similar to how the deliquescence at 50 %RH of chloride and bromide salts is reversed for $K^+$ and $Na^+$, $NaH_2PO_4$ is deliquescent and $KH_2PO_4$ is not (reversed relative to the dibasic phosphate salts used above). The condensation of glycine in the deliquescent $NaH_2PO_4$ exceeded 8% through ten cycles, while the yield in non-deliquescent $KH_2PO_4$ was 0% (see Supplementary Figs. 20–22 and Supplementary Tables 18, 19).

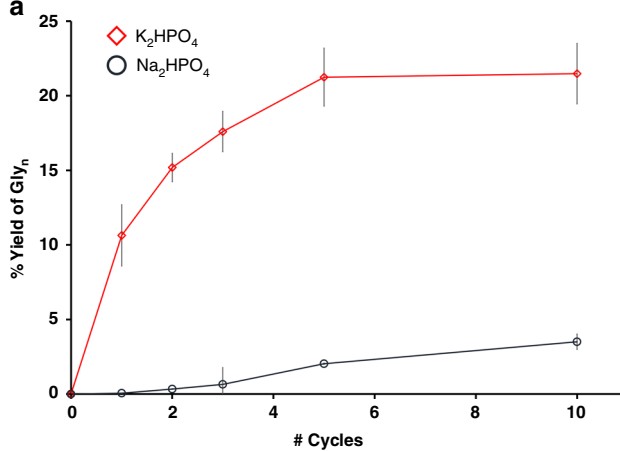

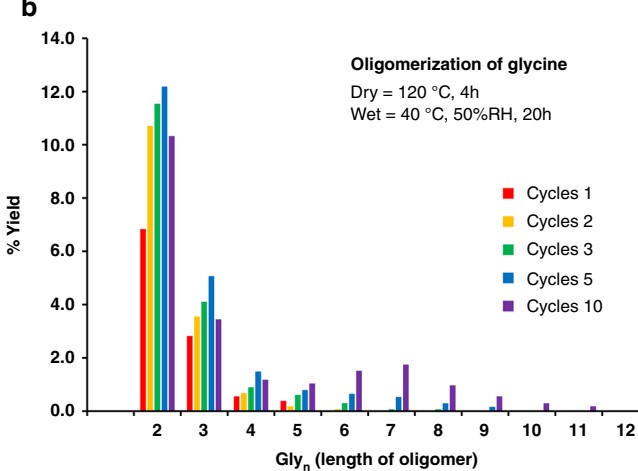

**Fig. 4** Glycine oligomerization in $K_2HPO_4$ vs. $Na_2HPO_4$. **a** The total yields of glycine oligomers (excluding DKP) in the presence of $K_2HPO_4$ (red diamonds) or $Na_2HPO_4$ (blue circles) after 1, 2, 3, 5, and 10 cycles. Each cycle was 24 total hours: 20 h at 40 °C and 50 %RH followed by 4 h at 120 °C. The error bars represent 95% confidence intervals ($n = 3$ identical experiments). **b** Distribution of yields by oligomer length after cycles 1, 2, 3, 5, and 10 in the sample prepared with $K_2HPO_4$. Source data are provided as a Source Data file

## Discussion

The results for the mixtures that were identical except for the presence of potassium or sodium counterions deserve further discussion. In organic chemistry, $K^+$ and $Na^+$ are generally viewed as unreactive spectator ions that have little or no significant, specific influence on chemical reactions. However, in the systems reported here, the presence of $K^+$ versus $Na^+$ has a profound influence on the yield of glycine oligomerization. The fact that $K_2HPO_4$ is deliquescent at 50 %RH, while $Na_2HPO_4$ is not, is responsible for a near tenfold difference in yield.

We briefly surveyed the difference in deliquescence for $K^+$ and $Na^+$ salts of a variety of counteranions—including halides, carboxylates, carbonates, and phosphates at 30, 50, and 70 %RH (see Supplementary Fig. 1 and Supplementary Table 3). It was striking that for several of the anions most relevant to biological polymers, $K^+$ salts are typically deliquescent while $Na^+$ are not. Phosphates are the primary anions associated with nucleic acids, while carboxylates are anions commonly found on proteins. All cells in biology enrich potassium from their external environments, and the vast majority of cells have higher intracellular concentrations of $K^+$ than $Na^+$[46–48]. While no definitive explanation is known

for life's universal enrichment of potassium, it is reasonable to consider $K^+$ salts may have had a beneficial influence on the regulation of moisture and the solubility of protobiopolymers during life's development. $K^+$ may have also supported higher yields in our systems by its modest effect speeding the hydrolysis of DKP to $Gly_2$ and slowing the hydrolysis of $Gly_2$ to Gly (relative to $Na^+$)[49]. We are aware of only one other report of a prebiotic system where swapping $K^+$ for $Na^+$ resulted in significant differences in yield[50].

We are particularly interested in what other substrates can undergo condensation to form prospective protobiopolymers, which includes expanding our studies to amino acids other than glycine. We are also interested in surveying other deliquescent minerals for their effects on yields in these systems, including sulfate and carbonate salts, and the influence of water activity[51]. Studies of the kinetics and thermodynamics of condensation and hydrolysis will be of critical importance in any thorough evaluation of these systems. The effects of the inclusion of prebiotically relevant catalysts, such as transition metals and clay minerals, are also of great interest to us. Some clay and salt minerals have been shown to catalyze the formation of potential protobiopolymers and protocellular compartments, including through condensation mediated by wet–dry cycling[10,52,53]. We reiterate that this communication is an initial report of a distinctive system. We will continue to test it, and we encourage others to do so as well.

Wet–dry cycles regulated by natural diurnal oscillations in temperature and humidity—not on uncontrollable rain events—represent an alternative model for driving the prebiotic formation of biopolymers by condensation reactions. Reaction mixtures with naturally deliquescent minerals regulate the extent of their hydration during wet–dry cycling. Above a critical relative humidity, these deliquescent reaction mixtures naturally collect a minimal amount of ambient water vapor to gain the advantages associated with dissolving into aqueous solution without gathering a large excess of water that would drive deleterious hydrolysis reactions. Rehydration mediated by deliquescence offers a model with improved prebiotic relevance, as it avoids the problems of overdilution expected in systems that rely on uncontrolled events, like rainstorms or flooding. The tendency for anions commonly found in modern biopolymers—like phosphates and carboxylates—to have lower critical humidities for deliquescence when paired with $K^+$ vs. $Na^+$ cations suggests life's preference for potassium may be due, in part, to the regulation of moisture content as life developed and evolved.

## Methods

**Cycling experiments**. A typical experiment entailed massing the solid salts and glycine, then thoroughly mixing the dry powders using an IKA M20 batch mill. The salts and glycine were mixed in a 2:1 molar ratio (the total moles of salt is twice the moles of glycine at the start of the reaction), and the total mass of the solid mixture stock was ~20 g. For a standard $(K^+/Na^+)(^-Cl/^-OH)$-glycine stock mixture, we massed 7.00 g of glycine (93 mmol), 3.23 g of KCl (47 mmol), 8.17 g of NaCl (139 mmol), 0.13 g of KOH (2.3 mmol), and 0.30 g of NaOH (6.9 mmol), and transferred the solids into the mill. The mixture was milled for 2 min and transferred to scintillation vials in 50 mg portions. Each reaction was run in triplicate, and each run required five vials per reaction mixture, as one vial was removed for analysis at the end of cycles 1, 2, 3, 5, and 10 (to avoid the problems posed by attempting to sample a potentially heterogeneous solid mixture). For the wet phase of each cycle, the vials were placed in a temperature and humidity-controlled chamber (an FWE Clymate IQ® cabinet) set to 40 °C and either 30, 50, or 70 %RH. We ensured the precision of these relative humidities by placing independent hygrometers (Fisherbrand™ Traceable™ Jumbo Thermo-Humidity Meter) in each of the chambers. Throughout our experiments, the recorded relative humidity stayed within ± 5%. For the dry phase of each cycle, the samples were transferred to a hot plate set to 100 °C or 120 °C for 6 or 4 h, respectively. As discussed briefly above and in the Supplementary Discussion, there is disagreement in the literature over prebiotic temperatures and length of day, but these conditions seem within the bounds of reason. The samples were analyzed at the conclusion of the drying

portion of cycles by IP–HPLC and MALDI–TOF MS (see Supplementary Figs. 29, 30 and Supplementary Table 24).

**Analytical methods**. We have described a method for the quantitative analysis of mixtures of oligoglycine by IP–HPLC[39]. The dry samples from cycling experiments were dissolved in 3.0 mL 0.1% TFA in $H_2O$ and filtered before analysis by HPLC. The analysis was performed on a Shimadzu LC-20AR fitted with a Phenomenex Luna® C18 column ($250 \times 4.6$ mm, 3 μm particle size). The mobile phase was an aqueous solution of 50 mM $KH_2PO_4$ and 7.2 mM $C_6H_{13}SO_3Na$, adjusted to pH 2.5 by the addition of HPLC-grade 85% $H_3PO_4$. The mobile phase was used iso-cratically with a flow rate of 1.00 mL min$^{-1}$. The column oven was maintained at 30 °C, and samples were injected in 5.0 μL aliquots by autosampler, with detection at 195 nm and 214 nm. We report individual yields for each oligomer based on the percent glycine converted to that product (see Supplementary Fig. 28 for chromatograms from a typical cycling experiment). The total yield is the sum of the yields of all $Gly_{n\geq2}$ products, not including the cyclic dimer of glycine, 2,5-diketopiperazine (DKP). DKP is generally viewed as a prebiotically disadvantageous side product that consumes amino acid building blocks to the detriment of the formation of longer oligomers[54,55]. As reported in previous oligomerizations by Cronin et al., our reaction mixtures generally yellowed over the course of the experiment and some produced insoluble white solids[14]. These solids—presumably, higher $Gly_n$ oligomers insoluble in water—were removed by filtration prior to analysis by HPLC.

**Statistics**. All experiments were performed in triplicate. Error bars in figures represent 95% confidence intervals (CIs) based on the $t$ critical values for two-tailed tests. For readability in the text, yields are reported as mean averages without the corresponding CIs.

## Data availability

The data that support the findings of this study are available from the corresponding author upon reasonable request. The raw data underlying Figs. 1–4, Supplementary Figs. 2–28 and 31, and Supplementary Tables 1, 2, and 4–23 are provided as a Source Data file.

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

## Acknowledgements

This work was jointly supported by the NSF and the NASA Astrobiology Program, under the NSF Center for Chemical Evolution (CHE-1504217). T.D.C. was supported by the NASA Earth & Space Science Fellowship Program (Award #80NSSC17K0521). The NSF and Saint Louis University jointly funded the spectrometer used to acquire ICP–OES data through the NSF Major Research Instrumentation Program (Award CHE-1626501). We are particularly grateful to Mathew T. Graham, Prof. Nicholas V. Hud, and Prof. Martha A. Grover for helpful discussions.

## Author contributions

P.J.B. conceived the project. T.D.C., R.F., and P.J.B. designed the wet–dry cycling experiments. T.D.C., R.F., J.T.M., and H.E.K. performed the wet–dry cycling experiments. T.D.C. and R.F. performed the HPLC analysis of samples. J.G.F. designed and conducted the MALDI–TOF experiments. All authors discussed the results and edited the manuscript prepared by T.D.C., R.F., J.G.F., and P.J.B.

## Additional information

**Competing interests:** The authors declare no competing interests.

**Peer Review Information**: *Nature Communications* thanks Alfonso Davila, Stefan Fox and other anonymous reviewer(s) for their contribution to the peer review of this work. Peer reviewer reports are available.

