## [Peer Review File · Nature Communications]

Reviewers' comments:

Reviewer #1 (Remarks to the Author):

Prebiotic Condensation Through Wet–Dry Cycling Regulated by Deliquescence
Campbell et al.

This paper proposes a theoretical scenario for the synthesis of prebiotic peptides based on wet/dry cycles driven by deliquescence of a salt substrate. This scenario is supported by laboratory experiments where different salts spiked with Glycine were controllably dried at high temperature (100–120°C) after they had reached the deliquescence point. The yields of oligomers were significantly higher in samples that were dried after deliquescence compared to samples that never reached the deliquescence point. The length of the produced oligomers also increased with the number of wet/dry cycles. The authors argue that salt deliquescence regulates the amount of liquid water available during wet cycles to levels where reactants can be dissolved, but where hydrolysis does not dominate over condensation. The authors further speculate that the enrichment of potassium typically observed in living cells could be due to the need of the first living systems to acquire and retain sufficient water to serve as a solvent for biochemical reactions.

The hypothesis that hygroscopic salts could have been a favorable substrate for prebiotic condensation of peptides is novel and significant, and appears to be supported by the experimental results. The results presented ought to be of interest to researchers in the prebiotic chemistry community and also to the wider scientific community, and will likely motivate similar experiments to validate the results presented, or to test the same hypothesis on other types of biological polymers. I consider that the paper is suitable for publication if the authors can address the issues outlined below:

The author's interpretation of how and why salt deliquescence might favor polymer condensation over hydrolysis is not entirely complete, as it appears to obviate the importance of low water activity solutions in polymer synthesis. The authors emphasize how deliquescence regulates moisture content (the amount of water that goes into the system), during wet cycles, as opposed to stochastic storms that might provide too much water and favor hydrolysis. But when salts deliquesce they form a saturated brine of low water activity, which ought to be more favorable for condensation reactions than pure water (i.e., rain). The water activity of such brines is often constant, even after changes in temperature, and might decrease as the brine approaches the efflorescence point, which should also favor condensation. In this scenario, it is not the amount of water in the system what favors condensation over hydrolysis, but the low activity of the solution, and the fact that it remains constant, or decreases, during the wet/dry cycle.

Of course, the amount of water in the system determines the water activity of the solution, so water activity is indirectly a part of the author's interpretation, but water activity is never mentioned in the manuscript. I think the authors can easily strengthen the paper by discussing the issue of water activity more directly. This is not only a technicality, because it implies that salts with lower deliquescence point, and therefore lower water activity, might be even more favorable for polymer synthesis than salts with higher deliquescence point, even though both substrates would tightly regulate hydration. This is also relevant for the interpretation of results in Figure 3. Adding 20 mL of (pure?) water (excess hydration) resulted in much lower yields, but this is likely due to the change in water activity of the resulting mixture, rather than the volume of water added. Adding a similar volume of a saturated brine might not affect the yields.

More minor comments:

Page 4. The statement that the volume of Don Juan Pond oscillates with humidity is misleading. It suggests that the increase in volume is due to deliquescence—i.e. direct transfer of water vapor into the pond at high RH—but this is not the case. The volume of the pond changes due to changes in runoff from different sources. One source is ice-melt, which increases with increasing insolation. Another source, explained in the Dickinson et al paper, are water tracks whose activity increases with RH and discharge salty water into the pond. That is where deliquescence plays a role in the hydrology of the system.

Page 15. The idea that life's preference for potassium may be due, in part, to the regulation of moisture content by K-bearing salts is too speculative. It presupposes that the first cellular entities with a protoplasm also evolved within deliquescent substrates, and relied on moisture cycles regulated by the substrate. This is not supported by the data presented in the study, and I recommend that it is removed.

Sincerely

Alfonso Davila
NASA Ames Research Center

Reviewer #2 (Remarks to the Author):

The question of how deliquescent salts interact with the polymerization process is very relevant and interesting to prebiotic chemistry and origin of life. This paper is novel and I find the results quite convincing. The majority of the paper is experimental methods, and I do not feel well-qualified as a reviewer to comment on this. I support publication of this paper, but I hope that you will also seek input from other reviewers.

Reviewer #3 (Remarks to the Author):

General Comments:

An experimental study dealing with the condensation of glycine under wet-dry cycles in the presence of a few minerals is evaluated for publication in Nature Communications. Such condensation reactions are considered important for prebiotic chemistry for creating polypeptides. The key finding is to provide a new scientific characterization of hydration/dehydration events under the proposed planetary conditions. The authors selected sodium and potassium salts to provide a simple but elegant comparison of how deliquescent minerals can (or cannot) facilitate the oligomerization of glycine. At elevated temperature (100 to 120 Celsius) evaporation to dryness of the precursor chemicals occurs in 4-6 h. Instead, at lower temperatures under 70% relative humidity, the spontaneously gain of water forms real aqueous solutions during these model cycling. As a result of this deliquescence process, an order of magnitude improvement in oligomerization is observed relative to controls. A remarkable aspect of the report is to highlight how selected salts of potassium are of fundamental importance to regulate moisture content to be above the limiting water amount needed for reaching the full dissolution of reactants to enable condensation, but to avoid a detrimental excess that favors hydrolysis. This is an interesting piece of work, generally enjoyable to read, and could attract considerable attention to the journal. Please note that there are a few instances where the discussion of some supporting references will improve the outcome of the manuscript. For example:

- 1) When stating that "Wet-dry cycling is commonly viewed as a feasible means of driving condensation reactions in prebiotic conditions to generate mixtures of prospective protobiotopolymers, including peptides, depsipeptides, nucleic acids, and others.6-15", the authors considered among other points how clays environments could contribute to the thermal condensation of amino acids (ref. 10). This should be better presented with a separate statements. In addition, it would be really important to expand this idea with the by explaining that clay minerals can also promote the production of vesicles (Hanczyc et al., Science. 2003 Oct 24;302(5645):618-22. DOI: 10.1126/science.1089904), and catalyze in their interlayers of variable widths important prebiotic reactions (Zhou et al., Sci. Rep. 2017 Apr 3;7(1):533. DOI: 10.1038/s41598-017-00558-1).
- 2) When explaining Figure 2b: Can the manuscript add an explanation about the fate of the Glycine oligomers? For example as the yield starts to decay in the figure for 70% RH. Ref.: Shock,

Geochimica et Cosmochimica Acta 1992, 56 (9), Sep, 3481-3491. DOI: 10.1016/0016-7037(92)90392-V.

Overall, this is a terrific paper that brings refreshing concepts to the field. Reviewer recommends to the editor that after adding these statements and references, the manuscript should be quickly published.

Reviewer #4 (Remarks to the Author):

Manuscript "Prebiotic Condensation Through Wet-Dry Cycling Regulated by Deliquescence" by Thomas D. Campbell et al.

Referee report:

In their manuscript, Campbell et al. address a limited but important problem in prebiotic chemistry. Referring to the conventional wet-dry cycling scenario, they rightly state that "One obvious shortcoming of the model is its reliance on rain storms or flooding as controlled sources of water." They present an alternative scenario in which the wet phase is caused by water uptake of deliquescent minerals. This is an intriguing idea. The authors have experimentally simulated this novel scenario by using the oligomerization of glycine as a model reaction. Wet-dry cycling between 40 °C and 100 or 120 °C (simulating diurnal temperature variations on the young Earth) in the presence of deliquescent salts produced oligoglycines in considerable yields. Also, the oligomers obtained were relatively long. The humidity and the nature of the salts were shown to be of crucial importance.

The results presented support the plausibility of wet-dry cycling as a prebiotic route to condensation products. The deliquescent-minerals scenario is an important extension of our understanding of prebiotic condensation reactions, though it will not make conventional wet-dry scenarios obsolete and may have its own weaknesses.

The manuscript is clearly written. Its length is appropriate to its contents. The conclusions drawn from the experiments are sound, and the experimental methods are carefully described. The HPLC method used looks reliable.

I recommend publication with some minor suggestions and corrections.

1. As far as I could see, no sulfates were studied. This seems surprising because many evaporates are sulfate minerals. Perhaps it would be appropriate to include sulfates in potential follow-up studies. In this context, I suggest to cite in the present manuscript the work by Hazen on Hadean minerals (R.M. Hazen (2013) Paleomineralogy of the Hadean eon: A preliminary species list. *American Journal of Science* 313, 807-843).

2. The following very recent paper on the formation of oligoglycines by wet-dry cycling with and without clay minerals should also be cited: Fox et al. (2019) An automated apparatus for the simulation of prebiotic wet-dry cycles under strictly anaerobic conditions. *International Journal of Astrobiology* 18, 60-72.

3. Page 7, line 3f: "Hydroxide was selected as a simple source of base to catalyze condensation, which will not proceed efficiently near neutral pH." Perhaps a word should be said about acidic pH to avoid the wrong impression that condensation is only efficient at alkaline pH. For example, sodium dihydrogenphosphate which gives acidic solutions gave a yield of 6.3 %. This is less than under basic conditions but still quite significant (Table S1).

4. Alkaline conditions were used in most of the experiments. However, the early Earth was mainly not alkaline (acidic volcanic gases, high partial pressure of carbon dioxide, ocean pH probably around 7). Therefore, a sentence may be added which describes where alkaline conditions could have existed (e.g., over basalt).

5. Page 13, line 9f: The statement that "longer oligomers are better suited to developing the secondary and tertiary structure observed in modern functional proteins" is undoubtedly true. Here, it could be added that even moderately long oligomers may have acted as catalysts in a protometabolism, despite being much smaller than proteins.

6. Page 5, line 18: There is no doubt that the days were shorter in the Hadean, because tidal forces must have slowed Earth's rotation over the last 4 Ga. Therefore, the word "speculation" should be substituted accordingly.

7. Page 6, line 20: "is the only amino acid". It would be more precise to write "is the only canonical amino acid" (because there are also achiral dialkyl amino acids).

8. Page 15, line 14ff: In this context, it may be interesting that during the evaporation of sea

water, first NaCl and then potassium salts precipitate. Thus, potassium accumulates in solution. 9. In some experiments, the total yield of glycine oligomers seems to decrease from cycle #5 to cycle #10 (Figures 2 and S15). Is the reason for this known? Perhaps formation of DKP as a "dead end" or decomposition of glycine?

In summary, the manuscript is an important contribution to prebiotic chemistry.

Stefan Fox

RESPONSES TO THE COMMENTS OF THE REFEREES

Reviewer #1

1-1. Reviewer #1: *“This paper proposes a theoretical scenario for the synthesis of prebiotic peptides based on wet/dry cycles driven by deliquescence of a salt substrate. This scenario is supported by laboratory experiments where different salts spiked with Glycine were controllably dried at high temperature (100-120°C) after they had reached the deliquescence point. The yields of oligomers were significantly higher in samples that were dried after deliquescence compared to samples that never reached the deliquescence point. The length of the produced oligomers also increased with the number of wet/dry cycles. The authors argue that salt deliquescence regulates the amount of liquid water available during wet cycles to levels where reactants can be dissolved, but where hydrolysis does not dominate over condensation. The authors further speculate that the enrichment of potassium typically observed in living cells could be due to the need of the first living systems to acquire and retain sufficient water to serve as a solvent for biochemical reactions.”*

“The hypothesis that hygroscopic salts could have been a favorable substrate for prebiotic condensation of peptides is novel and significant, and appears to be supported by the experimental results. The results presented ought to be of interest to researchers in the prebiotic chemistry community and also to the wider scientific community, and will likely motivate similar experiments to validate the results presented, or to test the same hypothesis on other types of biological polymers.”

Our Reply: We thank the referee for this review, which has been very helpful in improving our manuscript. We have grouped related comments together and addressed each grouped issue, below.

1-2. Reviewer #1: *“I consider that the paper is suitable for publication if the authors can address the issues outlined below:*

The author’s interpretation of how and why salt deliquescence might favor polymer condensation over hydrolysis is not entirely complete, as it appears to obviate the importance of low water activity solutions in polymer synthesis. The authors emphasize how deliquescence regulates moisture content (the amount of water that goes into the system), during wet cycles, as opposed to stochastic storms that might provide too much water and favor hydrolysis. But when salts deliquesce they form a saturated brine of low water activity, which ought to be more favorable for condensation reactions than pure water (i.e., rain). The water activity of such brines is often constant, even after changes in temperature, and might decrease as the brine approaches the efflorescence point, which should also favor condensation. In this scenario, it is not the amount of water in the system what favors condensation over hydrolysis, but the low activity of the solution, and the fact that it remains constant, or decreases, during the wet/dry cycle.

Of course, the amount of water in the system determines the water activity of the solution, so water activity is indirectly a part of the author’s interpretation, but water activity is never mentioned in the manuscript. I think the authors can easily strengthen the paper by discussing the issue of water activity more directly. This is not only a technicality, because it implies that salts with lower deliquescence point, and therefore lower water activity, might be even more favorable for polymer synthesis than salts with higher deliquescence point, even though both substrates would tightly regulate hydration.

This is also relevant for the interpretation of results in Figure 3. Adding 20 mL of (pure?) water (excess hydration) resulted in much lower yields, but this is likely due to the change in water activity of the resulting mixture, rather than the volume of water added. Adding a similar volume of a saturated brine might not affect the yields.”

Our Reply: We agree with the reviewer that low water activity is crucial to the concept of driving condensation reactions that would otherwise favor hydrolysis and that we should explicitly say so in the manuscript. During the dry-down phase, the activity of water must be very low, as the water present is driven away from the solid mixture. We have not yet undertaken a thorough analysis of the water activity in the wet phase, but analyzing what sorts of reactions the “wet” system supports will be the subject of future work from our group.

As a preliminary attempt to address the concerns of the reviewer, we performed two additional experiments described in the following text inserted to page S10 of the Supplementary Information:

Effect of Water Activity. We conducted two modified cycling experiments to test the effect of water activity in these systems. First, we performed a standard cycling experiment with the modification of not evaporating water from the samples during heating periods. We allowed the initial reaction mixture of glycine in the $(K^+, Na^+)(Cl^-, OH^-)$ salt mixture to acquire water over 5 days at 40° C and 70 %RH. We then placed this mixture in a capped tube and subjected it to cycled phases of 6 h at 100 °C and 18 h at 40 °C. The solution did not evaporate because it was capped, but we still observed the formation of some Gly₂ and Gly₃ despite the presence of water and the possibility of hydrolysis. The results are plotted in Supplementary Fig. 31, as a green line added to the plot that appears as Figure 3 in the main paper. This experiment suggests the water activity in the system is low enough to support the formation of some condensation product, but relative to the lower water activity afforded by evaporating to dryness, the yields and lengths of the oligomers are lower.

We also modified the “overhydration” experiment plotted in Figure 3 of the main paper by rehydrating with brine rather than deionized water. This experiment simulates an evaporating pond that is periodically refilled by ocean water. We expected to add 20 mL of brine after each cycle, but the amount of salt present led to precipitation during the first drying phase that precluded adding more brine after the first addition. A cap of precipitated salt formed at the top of the solution and prevented evaporation to dryness (see photograph in Supplementary Fig. 32). We continued to heat (6 h at 100 °C) and cool (6 h at 100 °C & 70 %RH) the samples for ten cycles, but we did not observe the formation of any oligomers by IP–HPLC. For this system, it appears the water activity is high enough to favor hydrolysis over condensation, though we hesitate to draw firm conclusions without further experimentation.

We have inserted the following sentence on p. 12 of the manuscript:

While the heat and reduced water activity during the evaporative dry phase support peptide synthesis, heat applied to an overhydrated sample—with higher water activity—can favor hydrolysis. The liquid deliquescent brine did have water activity low enough to support peptide condensation, but the evaporation of samples to dryness was critical to the growth of longer oligomers. Control samples of glycine in liquid deliquescent brines heated in capped vials—and hence, not allowed to evaporate—produced trace yields of Gly₂ and Gly₃ over 10 cycles (1.1% of

Gly₂₊₃), but the yields were comparatively lower than the samples allowed to evaporate to dryness (16.0% of Gly_{≥2}, with up to Gly₁₃ observed, see Supplementary Fig. 31).

We used ICP–OES to determine the concentrations of Na⁺ and K⁺ in the brine of the closed-capped experiment and added the following sentence to page S5 of the Supplementary Information:

The 3:1 ratio (of Na:K) was preserved throughout the cycling experiments, as determined by ICP–OES analysis of the “closed-capped” samples (Supplementary Table 25) in the experiment described on page S11.

We have also inserted the following table and two figures and their accompanying captions into the Supplementary Information of the revised submission:

	[Na ⁺] (ppm)	[Na ⁺] (M)	[K ⁺] (ppm)	[K ⁺] (M)	Na ⁺ :K ⁺
Cycle 1	100	4.3	58	1.5	2.9 : 1
Cycle 2	86	3.7	51	1.3	2.9 : 1
Cycle 3	91	4.0	53	1.4	2.9 : 1
Cycle 10	88	3.8	53	1.4	2.8 : 1

Supplementary Table 25. Ion concentrations obtained by ICP–OES analysis of the closed-capped samples for the (Na⁺,K⁺)(Cl⁻,OH⁻) system. Each cycle was 18 hours at 40 °C and 70 %RH followed by 6 hours at 100 °C.

Supplementary Figure 31. Yields of glycine oligomers (excluding DKP) in the presence of (K⁺,Na⁺)(Cl⁻,OH⁻) after 1, 2, 3, 5, and 10 cycles. Each cycle was 24 hours. For the samples marked

with triangles, one cycle included 18 hours at 40 °C and 70 %RH, followed by 6 hours at 100 °C. The samples marked with circles were exposed to the same environmental cycles, but 20 mL of water was added to the mixture before each drying period. This addition simulated heavy rain and overhydration of the sample to verify a shortcoming of the standard model for wet–dry cycling that is obviated by self-regulated, limited rehydration through deliquescence. The samples marked with crosses represent the closed-capped system, which tested the propensity for glycine oligomerization in samples not permitted to evaporate during the hot phase of each cycle. The error bars represent 95% confidence intervals based on three replicate experiments.

Supplementary Figure 32. Photograph of the reaction vials in the “overhydration with brine” experiment described on page S8. During evaporation, a crust of salt sealed the liquid reaction mixture, preventing evaporation to dryness during the hot phase of each cycle.

1-3. Reviewer #1: “*More minor comments: Page 4. The statement that the volume of Don Juan Pond oscillates with humidity is misleading. It suggests that the increase in volume is due to deliquescence—i.e. direct transfer of water vapor into the pond at high RH—but this is not the case. The volume of the pond changes due to changes in runoff from different sources. One source is ice-melt, which increases with increasing insolation. Another source, explained in the Dickinson et al paper, are water tracks whose activity increases with RH and discharge salty water into the pond. That is where deliquescence plays a role in the hydrology of the system.*”

Our Reply: We agree this passage could be misleading to readers as written and have modified the manuscript to read:

Old sentence (p. 4 of the manuscript)

“The Don Juan Pond in Antarctica is rich in deliquescent CaCl_2 , and the volume of the pond oscillates seasonally based on fluctuations in the local relative humidity.”

New Sentence (p. 4 of the manuscript):

“The Don Juan Pond in Antarctica is adjacent to several steep-sloped water tracks that are rich in deliquescent CaCl_2 . In summertime, meltwaters running through the tracks wash calcium-rich brine into the pond, augmenting water levels that fluctuate seasonally in a body of water that rarely freezes despite surface temperatures that typically descend to $-50\text{ }^\circ\text{C}$ in winters.”
(with citation to Dickson, 2013)

1-4. Reviewer #1: *“Page 15. The idea that life’s preference for potassium may be due, in part, to the regulation of moisture content by K-bearing salts is too speculative. It presupposes that the first cellular entities with a protoplasm also evolved within deliquescent substrates, and relied on moisture cycles regulated by the substrate. This is not supported by the data presented in the study, and I recommend that it is removed.”*

Our Reply: We understand and appreciate the reviewer’s concern about speculation with regard to the origin of life. Any attempt to provide answers to questions relating to life’s origins—roughly four billion years ago—is going to be inherently speculative. It would be impossible to make progress in this field if we abstained from speculation altogether.

Mindful of this delicateness, we have tried to be very careful with our diction and limited the extent of the most speculative aspects of our manuscript. The main speculative statement is “While no definitive explanation is known for life’s universal enrichment of potassium, it is reasonable to consider K^+ salts may have had a beneficial influence on the regulation of moisture and the solubility of protobiopolymers during life’s development. K^+ may have also supported higher yields in our systems by its modest effect speeding the hydrolysis of DKP to Gly_2 and slowing the hydrolysis of Gly_2 to Gly (relative to Na^+)” on page 15 of the manuscript. We note that we have not asserted this idea as a conclusion, but merely used a single sentence to suggest the idea is “reasonable to consider”. We are unaware of this idea having been raised before, so we think it is a useful contribution to the variety of ideas in the field.

The surrounding sentences in the section also provide data and other information that establishes a context for the idea to help to justify its merits. Namely, that potassium salts of many biologically relevant anions are more hygroscopic than their sodium counterparts. And, as a counterpoint, the subsequent sentence notes an alternate explanation for how K^+ might support the growth of longer polymers. We are not trying to champion the idea that deliquescence was important, so much as raise it as a possibility for others to consider and challenge with the scientific method. But the idea cannot be tested if it is not mentioned.

Thus, taken in whole, we believe that the statement on page 15 is carefully worded and placed in the appropriate context to stimulate the thoughts of a reader without misleading him or her into a flawed conclusion. Correspondingly, we do not believe this sentence in the manuscript should be modified.

Reviewer #2

2-1. Reviewer #2: *“The question of how deliquescent salts interact with the polymerization process is very relevant and interesting to prebiotic chemistry and origin of life. This paper is novel and I find the results quite convincing. The majority of the paper is experimental methods, and I do not feel well-qualified as a reviewer to comment on this. I support publication of this paper, but I hope that you will also seek input from other reviewers.”*

Our Reply: We thank the referee for these kind comments.

Reviewer #3

3-1. Reviewer #3: *“An experimental study dealing with the condensation of glycine under wet–dry cycles in the presence of a few minerals is evaluated for publication in Nature Communications. Such condensation reactions are considered important for prebiotic chemistry for creating polypeptides. The key finding is to provide a new scientific characterization of hydration/dehydration events under the proposed planetary conditions. The authors selected sodium and potassium salts to provide a simple but elegant comparison of how deliquescent minerals can (or cannot) facilitate the oligomerization of glycine. At elevated temperature (100 to 120 Celsius) evaporation to dryness of the precursor chemicals occurs in 4-6 h. Instead, at lower temperatures under 70% relative humidity, the spontaneously gain of water forms real aqueous solutions during these model cycling. As a result of this deliquescence process, an order of magnitude improvement in oligomerization is observed relative to controls. A remarkable aspect of the report is to highlight how selected salts of potassium are of fundamental importance to regulate moisture content to be above the limiting water amount needed for reaching the full dissolution of reactants to enable condensation, but to avoid a detrimental excess that favors hydrolysis. This is an interesting piece of work, generally enjoyable to read, and could attract considerable attention to the journal.”*

Our Reply: We thank the referee for his/her kind comments.

3-2. Reviewer #3: *“Please note that there are a few instances where the discussion of some supporting references will improve the outcome of the manuscript. For example:*

1) When stating that “Wet–dry cycling is commonly viewed as a feasible means of driving condensation reactions in prebiotic conditions to generate mixtures of prospective protobiopolymers, including peptides, depsipeptides, nucleic acids, and others.⁶⁻¹⁵”, the authors considered among other points how clays environments could contribute to the thermal condensation of amino acids (ref. 10). This should be better presented with a separate statement. In addition, it would be really important to expand this idea with the by explaining that clay minerals can also promote the production of vesicles (Hanczyc et al., Science. 2003 Oct 24;302(5645):618-22. DOI: 10.1126/science.1089904), and catalyze in their interlayers of variable widths important prebiotic reactions (Zhou et al., Sci. Rep. 2017 Apr 3;7(1):533. DOI: 10.1038/s41598-017-00558-1).”

Our Reply: We agree that clays are also minerals of particular interest to the evolution of molecular complexity on the Prebiotic Earth. To address clays and some other topics raised below, we have

added the following section captioned Future Directions to page S15 of our Supplementary Discussion section:

The effects of the inclusion of prebiotically relevant catalysts, such as transition metals and clay minerals, is also of great interest to us. Some clay and salt minerals have been shown to catalyze the formation of potential protobiopolymers and protocellular compartments, including through condensation mediated by wet–dry cycling. (Lahav 1978, Hanczyc 2003, Zhou 2017)

We think it best for this sentence not to be inserted in the Introduction after the third sentence of the manuscript because (i) we do not specifically explore the use of clays in our manuscript and (ii) the Introduction focuses specifically on wet–dry cycling and dry-down reactions. Inserting the above sentence out of context might mislead or confuse the reader. In the Supplemental Discussion, it can appear within the proper context.

3-3. Reviewer #3: “2) When explaining Figure 2b: Can the manuscript add an explanation about the fate of the Glycine oligomers? For example as the yield starts to decay in the figure for 70% RH. Ref.: Shock, *Geochimica et Cosmochimica Acta* 1992, 56 (9), Sep, 3481-3491. DOI: 10.1016/0016-7037(92)90392-V.”

Our Reply: We have inserted the following sentence to the caption of Fig. 2

The downturn in total yield observed at higher temperature is addressed in the Supplementary Discussion.

We have added the following to the Supplementary Information on p. S9

Addressing the Downturn in Total Yield at Higher Temperatures. We speculate that the downturn in total yield sometimes observed at higher temperatures at higher cycles—such as between the fifth and tenth cycles at 120 °C in Figure 2b—could be attributable to the production of oligomers longer than Gly₁₄, diketopiperazine (DKP), or unknown decomposition products. Consistent with previous reports, we sometimes noted the formation of white precipitate—presumably insoluble oligomers of Gly_{>14}—that could not be analysed by our IP–HPLC method (Rodriguez-Garcia 2015). The apparent downturn could be due to our inability to measure products longer than Gly₁₄ rather than a decreased conversion of glycine to oligomer products. Alternate possible explanations for the downturn include thermal degradation of glycine and its oligomers (Shock 1992, Weiss 2018), as the reaction mixtures had a tendency to discolor (from white to pale yellow) with increasing time and heat, again consistent with previous reports of heated mixtures of glycine (Rodriguez-Garcia 2015). Finally, it is possible that the conditions favored the formation of DKP—cyclic Gly₂—which also could not be measured by our IP–HPLC method.

3-4. Reviewer #3: “Overall, this is a terrific paper that brings refreshing concepts to the field. Reviewer recommends to the editor that after adding these statements and references, the manuscript should be quickly published.”

Our Reply: We share this enthusiasm and thank the referee for his/her kind comments.

Reviewer #4

4-1. Reviewer #4: *“In their manuscript, Campbell et al. address a limited but important problem in prebiotic chemistry. Referring to the conventional wet–dry cycling scenario, they rightly state that “One obvious shortcoming of the model is its reliance on rain storms or flooding as controlled sources of water.” They present an alternative scenario in which the wet phase is caused by water uptake of deliquescent minerals. This is an intriguing idea. The authors have experimentally simulated this novel scenario by using the oligomerization of glycine as a model reaction. Wet–dry cycling between 40 °C and 100 or 120 °C (simulating diurnal temperature variations on the young Earth) in the presence of deliquescent salts produced oligoglycines in considerable yields. Also, the oligomers obtained were relatively long. The humidity and the nature of the salts were shown to be of crucial importance. The results presented support the plausibility of wet–dry cycling as a prebiotic route to condensation products. The deliquescent-minerals scenario is an important extension of our understanding of prebiotic condensation reactions, though it will not make conventional wet–dry scenarios obsolete and may have its own weaknesses.*

The manuscript is clearly written. Its length is appropriate to its contents. The conclusions drawn from the experiments are sound, and the experimental methods are carefully described. The HPLC method used looks reliable.

I recommend publication with some minor suggestions and corrections.”

Our Reply: We thank the referee for his comments.

4-2. Reviewer #4: *“1. As far as I could see, no sulfates were studied. This seems surprising because many evaporates are sulfate minerals. Perhaps it would be appropriate to include sulfates in potential follow-up studies. In this context, I suggest to cite in the present manuscript the work by Hazen on Hadean minerals (R.M. Hazen (2013) Paleomineralogy of the Hadean eon: A preliminary species list. American Journal of Science 313, 807–843).”*

Our Reply: We are currently pursuing other salts in our deliquescent model for a follow-up study. Sulfates will be included in this study. We did not include them in this study due to their perceived scarcity on the Prebiotic Earth (see: K. S. Habicht, M. Gade, B. Thamdrup, P. Berg, D. E. Canfield, *Science* 2002, 298, 2372), except perhaps for volcanic environments. We address the possibilities of expanding this study in the Future Directions section in the Supplementary Information, specifically with the insertion of the following text on p. S15:

We are also interested in surveying other deliquescent minerals for their effects on yields in these systems, including sulfate and carbonate salts.(ref: Hazen 2013)

4-3. Reviewer #4: *“2. The following very recent paper on the formation of oligoglycines by wet–dry cycling with and without clay minerals should also be cited: Fox et al. (2019) An automated apparatus for the simulation of prebiotic wet–dry cycles under strictly anaerobic conditions. International Journal of Astrobiology 18, 60–72.”*

Our Reply: We thank the reviewer for calling our attention to this manuscript. We have cited it as reference 16 in the sentence about wet–dry cycling on page 3 of the main paper:

Wet–dry cycling is commonly viewed as a feasible means of driving condensation reactions in prebiotic conditions to generate mixtures of prospective protobiopolymers, including peptides, depsi-peptides, nucleic acids, and others.

4-4. Reviewer #4: “3. Page 7, line 3f: “Hydroxide was selected as a simple source of base to catalyze condensation, which will not proceed efficiently near neutral pH.” Perhaps a word should be said about acidic pH to avoid the wrong impression that condensation is only efficient at alkaline pH. For example, sodium dihydrogenphosphate which gives acidic solutions gave a yield of 6.3 %. This is less than under basic conditions but still quite significant (Table S1).

4. Alkaline conditions were used in most of the experiments. However, the early Earth was mainly not alkaline (acidic volcanic gases, high partial pressure of carbon dioxide, ocean pH probably around 7). Therefore, a sentence may be added which describes where alkaline conditions could have existed (e.g., over basalt).”

Our Reply: We have changed a sentence in the main paper on page 7 from “which will not proceed efficiently near neutral pH” to “which will proceed less efficiently near neutral pH”

We have a brief discussion of pH/proton activity on page S12 of the Supplementary Information. We have added the following sentence to that section:

Still, peptide formation mediated by deliquescent wet–dry cycling does occur under mildly acidic conditions, which is notable considering the presumed increased acidity of the Prebiotic Earth (ref: Krissansen-Totton 2018).

Regarding alkaline conditions, we have inserted the following sentence on p. S13:

Alkaline conditions and hydroxide/oxide minerals have been proposed to exist on the early Earth in hydrothermal environments and those subjected to metamorphism and other geological processes. (ref: Hazen 2013, Russell 2014)

4-6. Reviewer #4: “5. Page 13, line 9f: The statement that “longer oligomers are better suited to developing the secondary and tertiary structure observed in modern functional proteins“ is undoubtedly true. Here, it could be added that even moderately long oligomers may have acted as catalysts in a protometabolism, despite being much smaller than proteins.”

Our Reply: We have inserted a sentence on p. 15 of the manuscript:

Even shorter peptides may have served as functional biomolecules on Prebiotic Earth. (ref: Carny 2005, Maury 2018, Wiczonek 2017)

4-7. Reviewer #4: *“6. Page 5, line 18: There is no doubt that the days were shorter in the Hadean, because tidal forces must have slowed Earth’s rotation over the last 4 Ga. Therefore, the word “speculation” should be substituted accordingly.”*

Our Reply: We have removed the words “there is speculation that” such that the revised sentence on p. 5 of the manuscript reads:

Although an Earth day was shorter at 4 Ga,²⁸ we selected cycles that lasted 24 hours as a reasonable approximation that was also convenient to monitor over many days.

4-8. Reviewer #4: *“7. Page 6, line 20: “is the only amino acid”. It would be more precise to write “is the only canonical amino acid” (because there are also achiral dialkyl amino acids).”*

Our Reply: We agree. We have inserted the word “canonical” such that the revised sentence on p. 6 reads:

Glycine is the only canonical amino acid that produces achiral oligomers, which simplifies their analysis relative to other amino acids.

4-9. Reviewer #4: *“8. Page 15, line 14ff: In this context, it may be interesting that during the evaporation of sea water, first NaCl and then potassium salts precipitate. Thus, potassium accumulates in solution.”*

Our Reply: The reviewer raises an interesting point that is actually the subject of a related project in our laboratory. In these experiments, we are looking at how KCl and NaCl can be separated under prebiotically relevant geological conditions of simulated prebiotic oceans. Evaporation is one of the main processes we are studying. We will publish these results within the next year.

4-10. Reviewer #4: *“9. In some experiments, the total yield of glycine oligomers seems to decrease from cycle #5 to cycle #10 (Figures 2 and S15). Is the reason for this known? Perhaps formation of DKP as a “dead end” or decomposition of glycine?”*

Our Reply: Reviewer #3 had a similar observation, which we have addressed above in point 3-3.

4-11. Reviewer #4: *“In summary, the manuscript is an important contribution to prebiotic chemistry.”*

Our Reply: We thank the reviewer for the kind words and the helpful comments above.

OTHER MODIFICATIONS TO THE ORIGINAL MANUSCRIPT

A-1. In the course of reading the literature, we discovered another a relevant reference to the topic of wet–dry cycles and how the wet phase both supports diffusion and enables hydrolysis. This reference predates our current citation of Higgs (2016). The Higgs paper goes into greater analysis of the issue, but it was an oversight not to have recognized the seminal contribution of Walker, et al. (2012). As such, in the two instances where we cited the Higgs paper, we have added a citation to Walker, et al. (on pages 3 and 11 of the manuscript).

A-2. The following additions to the ESI concerns the experiment performed in response to the first reviewers comments about water activity:

Selection of the Composition of the Deliquescent Chloride/Hydroxide Mixtures. **Add:** The 3:1 ratio was preserved throughout the cycling as shown by the ICP–OES measurements of the closed-capped samples (Table S25).

Procedure for Excess Rehydration Experiments. The (K/Na)(Cl/OH)-glycine stock mixture described in the main paper was transferred to scintillation vials in 50 mg portions. Each reaction was run in triplicate, and each run required five vials per reaction mixture, as one vial was removed for analysis at the end of cycles 1, 2, 3, 5, and 10 similar to the standard experiments. For the wet phase of each cycle, the vials were placed in a temperature and humidity controlled chamber (an FWE Clymate IQ® cabinet) set to 40 °C and 70 %RH. For the dry phase of each cycle, the samples were transferred to a hot plate set at 100 °C for 6 hours. As opposed to the standard experiments, 20 mL of water was added to the mixture before each drying period to simulate heavy rain and overhydration. The samples were analyzed at the conclusion of the drying portion of cycles by IP–HPLC.

Procedure for Closed-Capped, Excess Rehydration by Salt Brine Experiments. To assess the whether the water activity affects our deliquescent systems, a 5 g portion of the (K/Na)(Cl/OH)-glycine stock mixture described in the main paper was transferred to a high-pressure glass vial (Chemglass Life Sciences LLC, Vineland, NJ). The sample was let to deliquesce for 5 days in a humidity-controlled chamber set to 40 °C and 70 %RH (~7.5 g of water was sequestered at equilibrium, as determined by the increase in mass of the sample). The sample was capped and subjected to cycles of 100 °C for 6 hours while immersed in a sand bath followed by 18 hours at 40 °C. At the conclusion of hot phases for cycles 1, 2, 3, 5, and 10, 100 µL aliquots were transferred to microcentrifuge tubes for analysis by IP–HPLC and ICP–OES.

Density Measurements of the Deliquescent (K/Na)(Cl/OH)-glycine Brine. At the conclusion of the cycling experiment of the closed-capped system, three 800 µL aliquots of the sample were massed on an analytical balance (a Sartorius Practum 124–1S) to measure the density of the brine. The density was determined to be 1.33 ± 0.02 g/mL.

Procedure for Excess Hydration with Brine. As a control, the (K/Na)(Cl/OH)-glycine stock mixture described in the main paper was transferred to scintillation vials in 50 mg portions. The reaction was run in triplicate, and each run required five vials per reaction mixture, as one vial was removed for analysis at the end of cycles 1, 2, 3, 5, and 10, similar to the standard experiments. As

opposed to the experiments described above, 20 mL of a solution saturated in the $(\text{K}^+, \text{Na}^+)(\text{Cl}^-, \text{OH}^-)$ salt mixture was added to the reaction before the first drying period. For the wet phase of each cycle, the vials were placed in a temperature and humidity-controlled chamber (an FWE Clymate IQ® cabinet) set to 40 °C and 70 %RH. For the dry phase of each cycle, the samples were transferred to a hot plate set at 100 °C for 6 hours. At the conclusion of the last dry cycle, we removed the excess water from the system by vacuum evaporation using a Labconco CentriVap® vacuum concentrator at 45 °C under 5 Torr overnight. The samples were analyzed at the conclusion of the drying portion of cycles by IP-HPLC.

Inductively-Coupled Plasma–Optical-Emission Spectroscopy (ICP–OES). A 10 µL aliquot of each sample was diluted to 10 mL with 2% HNO_3 . Samples were measured for sodium and potassium concentrations using Perkin–Elmer Optima 8300 ICP Spectrometer. The instrument is equipped with Elemental Scientific (ESI) prepFAST autosampler system and utilizes a cross-flow nebulizer with the following parameters: plasma 10 $\text{Ar}_{(\text{g})} \text{min}^{-1}$; auxiliary 0.2 $\text{Ar}_{(\text{g})} \text{min}^{-1}$; nebulizer 0.65 $\text{Ar}_{(\text{g})} \text{min}^{-1}$; power 1,400 W. Calibration curves constructed for Na and K using a 10 mg L^{-1} elemental solutions (Sigma–Aldrich) resulted in $\leq 2\%$ error.

A-3. We have added this section to the Supplementary Discussion. The following text was added to page S15.

Future Directions. This paper is a communication of our initial results with respect to deliquescent wet–dry cycling. We have a number of follow-up experiments currently planned and in progress. We are particularly interested in what other substrates can undergo condensation to form prospective protobiopolymers, which includes expanding our studies to amino acids other than glycine. We are also interested in surveying other deliquescent minerals for their effects on yields in these systems, including salts of sulfates, carbonates, and bromides.

We remain interested in investigating the effects of water activity and what chemistry is supported at various points along these cycles. Studies of the kinetics and thermodynamics of condensation and hydrolysis will be of critical importance in any thorough evaluation of these systems. The effects of the inclusion of prebiotically relevant catalysts, such as transition metals and clay minerals, is also of great interest to us. Some clay and salt minerals have been shown to catalyze the formation of potential protobiopolymers and protocellular compartments, including through condensation mediated by wet–dry cycling. (Lahav 1978, Hanczyc 2003, Zhou 2017) We reiterate that this communication is an initial report of a novel system. We will continue to test it, and we encourage others to do so as well.

A-4. We have made a number of minor changes to align our manuscript with journal guidelines listed in the Manuscript Checklist for *Nature Communications*. These changes are tracked in the Word document and include: inserting full postal addresses, replacing all citations of figures in the ESI from “see Figure S##” to “see Supplementary Figure ##”, moving figure captions to appear below their corresponding figure, among other similar changes.

REVIEWERS' COMMENTS:

Reviewer #1 (Remarks to the Author):

I commend the authors for their efforts to address my comments. My original concerns with the manuscript have been resolved and I consider that the paper is suitable for publication.

Reviewer #3 (Remarks to the Author):

In order to satisfactorily address all comments from the previous revision, the manuscript should incorporate in page 16 or 17 the following text and references from the supporting supplement into the final version of the manuscript to be published:

"The effects of the inclusion of prebiotically relevant catalysts, such as transition metals and clay minerals, is also of great interest to us. Some clay and salt minerals have been shown to catalyze the formation of potential protobiopolymers and protocellular compartments, including through condensation mediated by wet–dry cycling.^{12, 17, 18} We reiterate that this communication is an initial report of a novel system. We will continue to test it, and we encourage others to do so as well."

In this way, the article will serve as an innovative source of inspiration and leadership in the origin of life field.

Reviewer #4 (Remarks to the Author):

In my opinion, with the changes made, this work is very valuable and should be published.

Kind regards,
Stefan Fox

RESPONSES TO THE COMMENTS OF THE REFEREES

Reviewer #1

1-1. Reviewer #1: *“I commend the authors for their efforts to address my comments. My original concerns with the manuscript have been resolved and I consider that the paper is suitable for publication.”*

Our Reply: We thank the referee again for the previous review.

Reviewer #3

3-1. Reviewer #3: *“In order to satisfactorily address all comments from the previous revision, the manuscript should incorporate in page 16 or 17 the following text and references from the supporting supplement into the final version of the manuscript to be published:*

“The effects of the inclusion of prebiotically relevant catalysts, such as transition metals and clay minerals, is also of great interest to us. Some clay and salt minerals have been shown to catalyze the formation of potential protobiopolymers and protocellular compartments, including through condensation mediated by wet–dry cycling.^{12, 17, 18} We reiterate that this communication is an initial report of a novel system. We will continue to test it, and we encourage others to do so as well.”

In this way, the article will serve as an innovative source of inspiration and leadership in the origin of life field.”

Our Reply: We thank the referee for his/her kind comments. We copied the sentences the reviewer identified in the Supplementary Discussion and the previous two sentences, for context, then inserted them into the main paper, where they span pages 16 and 17 of the revised manuscript. We then polished the text to read:

We are particularly interested in what other substrates can undergo condensation to form prospective protobiopolymers, which includes expanding our studies to amino acids other than glycine. We are also interested in surveying other deliquescent minerals for their effects on yields in these systems, including sulfate and carbonate salts, and the influence of water activity.⁵¹ Studies of the kinetics and thermodynamics of condensation and hydrolysis will be of critical importance in any thorough evaluation of these systems. The effects of the inclusion of prebiotically relevant catalysts, such as transition metals and clay minerals, is also of great interest to us. Some clay and salt minerals have been shown to catalyze the formation of potential protobiopolymers and protocellular compartments, including through condensation mediated by wet–dry cycling.^{10, 52, 53} We reiterate that this communication is an initial report of a distinctive system. We will continue to test it, and we encourage others to do so as well.

Reviewer #4

4-1. Reviewer #4: *“In my opinion, with the changes made, this work is very valuable and should be published.”*

Our Reply: We thank the referee for these kind comments and the previous review.

ADDITIONAL MODIFICATIONS TO THE REVISED MANUSCRIPT

A-1. In a recent search of the literature, we came across the following reference that discusses the discoloration that occurs during glycine polymerizations:

Ohara, S., Kakegawa, T., Nakazawa, H. Pressure effects on the abiotic polymerization of glycine. *Orig. Life Evol. Biosph.*, **37**, 215-223 (2007).

We have added this as supplementary reference 5 and added a citation to the following sentence on page 43 of the Supplementary Information. (The wording was already present in the SI, we just added a citation to the Ohara paper at the end of the last sentence.)

The apparent downturn could be due to our inability to measure products longer than Gly₁₄ rather than a decreased conversion of glycine to oligomer products. Alternate possible explanations for the downturn include thermal degradation of glycine and its oligomers,^{2, 4} as the reaction mixtures had a tendency to discolor (from white to pale yellow) with increasing time and heat, again consistent with previous reports of heated mixtures of glycine.^{2, 5}

A-2. On page 7, reworded the following sentence for clarity and readability:

Chromatographic analysis is crucial because the high quantity of salts present greatly complicates quantitative analysis by mass spectrometry

A-3. On page 8, we changed the legend’s description of panel B to avoid the passive voice (“A demonstration of...” instead of “were demonstrated”).

A-4. On page 11, we deleted the words “(aqua plot)” from the following sentence because it can confuse the reader into thinking it is some type of water rather than the color of the line in Figure 3. The line is sufficiently labelled in the legend to Figure 3 such that this extra information is unnecessary in the text.

In an experiment designed to study the effects of overhydration, the same mixtures of glycine with K^+ , Na^+ , Cl^- , and OH^- salts were subjected to phases of 6 h at 100 °C and 18 h at 40 °C & 70 %RH, but at the end of the cooling period, an additional 20 mL of deionized water (~~aqua plot~~) was added to a selection of the samples to simulate overhydration.

A-5. On page 11, we changed “did have” to “already had” to improve readability and clarity such that the sentence reads:

The liquid deliquescent brine already had water activity low enough to support peptide condensation, but the evaporation of samples to dryness was critical to the growth of longer oligomers.

A-6. On page 15, we added “that were” to the first sentence of the Discussion section, so it now reads:

The results for the mixtures that were identical except for the presence of potassium or sodium counterions deserve further discussion.